# Neurosyphilis: Still prevalent and overlooked in an at risk population

**Prashanth S. Ramachandran**[1]*, **Rob W. Baird**[2], **Peter Markey**[1,3], **Sally Singleton**[3], **Michael Lowe**[1], **Bart J. Currie**[1,4], **James N. Burrow**[1], **Ric N. Price**[1,4,5]*

**1** Division of Medicine, Royal Darwin Hospital, Darwin, NT, Australia, **2** Territory Pathology, Department of Health, Darwin, NT, Australia, **3** Centre for Disease Control, Darwin, Northern Territory, Australia, **4** Global and Tropical Health Division, Menzies School of Health Research, Charles Darwin University, Darwin, NT, Australia, **5** Centre for Tropical Medicine and Global Health, Nuffield Department of Medicine, University of Oxford, Oxford, United Kingdom

* Ric.Price@menzies.edu.au (RNP); prashanth.ramachandran@ucsf.edu (PSR)

**Data Availability Statement:** All relevant data are within the paper and its Supporting Information files.

## Abstract

### Background

Neurosyphilis (NS) presents with a variety of clinical syndromes that can be attributed to other aetiologies due to difficulties in its diagnosis. We reviewed all cases of NS from the "Top End" of the Australian Northern Territory over a ten-year period to assess incidence, clinical and laboratory manifestations.

### Methods

Patient data (2007–2016) were extracted from hospital records, centralised laboratory data and Northern Territory Centre for Disease Control records. Clinical records of patients with clinically suspected NS were reviewed. A diagnosis of NS was made based on the 2014 US CDC criteria. Results were also recategorized based on the 2018 US CDC criteria.

### Results

The population of the "Top End" is 185,570, of whom 26.2% are Indigenous. A positive TPPA was recorded in 3126 individuals. A total of 75 (2.4%) of TPPA positive patients had a lumbar puncture (LP), of whom 25 (35%) were diagnosed with NS (9 definite, 16 probable). Dementia was the most common manifestation (58.3%), followed by epilepsy (16.7%), psychosis (12.5%), tabes dorsalis (12.5%) and meningovascular syphilis (8.3%). 63% of probable NS cases were not treated appropriately due to a negative CSF VDRL. Despite increased specificity of the 2018 US CDC criteria, 70% of patient in the probable NS group were not treated appropriately. The overall annual incidence [95%CI] of NS was 2.47[1.28–4.31] per 100 000py in the Indigenous population and 0.95[0.50–1.62] in the non-Indigenous population (rate ratio = 2.60 [1.19–5.70];p = 0.017).

### Conclusion

Neurosyphilis is frequently reported in the NT, particularly in Indigenous populations. Disturbingly, 60% of probable neurosyphilis patients based on the 2014 criteria, and 70%

**Funding:** This study was supported by the Wellcome Trust, through a Senior Fellow in Clinical Science awarded to RNP (200909).

**Competing interests:** The authors have declared that no competing interests exist.

based on the 2018 criteria with were not treated appropriately. It is critical that clinicians should be aware of the diagnosis of NS and treat patients appropriately.

## Introduction

Syphilis continues to cause a major burden of disease due to its systemic manifestations and long-term neurological sequelae. After the introduction of contact tracing, followed by penicillin, the incidence of syphilis declined significantly, falling from 447 per 100 000-person years (py) in the United Stated in 1947 to 11.2 per 100 000py in 2000 [1, 2]. However since the turn of the millennium, the global incidence has increased markedly, more than doubling in at risk populations in North America, Australia and Europe [1, 3, 4]. In 2014 alone there were an estimated 6 million new cases of syphilis [5] the majority occurring in Africa and South East Asia [6]. The risk of syphilis is expected to be particularly high in certain ethnic groups, indigenous populations and men having sex with men [7].

Many countries have instituted mandatory reporting of all new cases of syphilis, to ensure early detection of outbreaks and to guide patient management and public health interventions. However, neurosyphilis (NS) is not a reportable disease as a separate entity and therefore epidemiological data are sparse. In the antibiotic era, several studies have estimated the incidence of NS to range from 0.08 to 2.2 per 100 000py [8–11]. However, the true incidence of NS is difficult to quantify due to frequent misdiagnoses, arising from a paucity of accurate microbiological tests, protean manifestations of the disease and varied clinical diagnostic criteria [12, 13].

In the tropical northern region of the Northern Territory (NT) of Australia (the "Top End"), which has had a centralised syphilis register and programmatic support since 2005, syphilis rates had been decreasing until 2013. However, in 2011 an epidemic of syphilis commenced in the State of Queensland which soon spread eastward to Central Australia, then the Top End of the Northern Territory by mid 2013 and subsequently to Western Australia. The tri-State epidemic continues in 2020. To quantify the impact of NS on the region, we undertook a review of all cases of NS from the "Top End" over a ten-year period (2007–2016) to assess its incidence and associated clinical and laboratory manifestations.

## Methods

### Study design

We conducted a retrospective (2007–2015) and prospective (2016) multicentre cohort study of NS, in the Top End of the Northern Territory of Australia. In 2007 the population of the 'Top End' was 168,036 rising to 194,882 in 2016. The corresponding Indigenous population rose from 45,765 (27.2% of population), to 52,101 (26.7% of population). The population at the midpoint of the ten-year study period was 185,570, of whom 48,632 (26.2%) were Indigenous. The midpoint population over the ten-year period was used for incidence calculations.

Between 2007 and 2016, all patients presenting to one of the three district hospitals servicing this region were eligible for inclusion in the study if they fulfilled the US Centres for Disease Control (US CDC) criteria for NS, comprising of a clinical syndrome consistent with NS and cerebrospinal fluid (CSF) changes [14]. Our initial analysis was based on the 2014 US CDC criteria for NS, which has a high sensitivity, allowing the capture of all cases, however, these criteria lack specificity, most notably when serological tests are negative [15]. The revised 2018 US CDC NS criteria were published during manuscript preparation and were included into subsequent epidemiological analysis (Table 1) [16].

**Table 1. Diagnostic criteria for neurosyphilis.**

|  | US CDC 2014 Diagnostic Criteria for NS | US CDC 2018 Diagnostic Criteria for NS |
|---|---|---|
| **Definite NS** | Serum TPPA positive | Serum TPPA positive |
|  | CSF VDRL positive | **Serum reactive RPR** |
|  |  | CSF VDRL positive |
| **Probable NS** |  | **Serum reactive RPR** |
|  | Serum TPPA positive | Serum TPPA positive |
|  | CSF WCC > 5 cell/microL | CSF WCC > 5 cell/microL |
|  | or | or |
|  | CSF protein > 0.5 mg/L | CSF protein > 0.5 mg/L |

Clinical syndromes consistent with NS were defined as meningitis with or without cranial nerve abnormalities, meningovascular disease, dementia (general paresis) and tabes dorsalis, and isolated presentations of epilepsy, psychosis, ocular and otic disease. Late manifestations of NS were categorized as dementia, tabes dorsalis and psychosis. Early manifestations of NS were categorized as meningitis, meningovascular disease, ocular disease, otic disease and isolated epilepsy. As per the US CDC 2014 NS criteria, a definite diagnosis of NS was defined as at least one clinical syndrome compatible with NS with a positive serum Treponema pallidum particle agglutination assay (TPPA) test and a positive CSF Venereal Disease Research Laboratory (VDRL). Probable NS was defined as a clinical syndrome suggestive of NS, a positive serum TPPA test, an elevated CSF white cell count (WCC) >5 cell/microL and/or CSF protein >0.45mg/L with no other cause found (Table 2). Possible NS was defined as a patient who was treated for NS due to high suspicion, but who did not meet CDC criteria for lack of a CSF examination, or a diagnosis based on an alternate criterion.

CSF treponemal tests (TPPA, FTA-Ab) is not included in either the 2014 or 2018 CDC criteria. The test carries a high sensitivity but low specificity, and is often used to rule out NS when negative. However due to its poor negative predictive value, some argue that it should not be used when a patient's pre-test probability of NS is high [17]. We did not use either CSF TPPA or FTA-Ab in our selection criteria as it was not part of the US CDC 2014 criteria.

## Data sources

Patients were identified and matched using three separate databases; i) The Northern Territory (NT) Centre for Disease Control notifications of syphilis, including classification (either less than 2 years duration or greater than 2 years or unknown duration) and register of prior treatment; ii) Hospital discharge coding from each of the three district hospitals; and iii) laboratory records of syphilis serology testing.

Demographic and clinical information from all eligible patients were gathered from the electronic medical records and the clinical classification of NS, its treatment and outcomes recorded in a standardised database using Excel (v16.19, Microsoft). Patients' hardcopy medical records were retrieved when further information was required. Regional population data were obtained from the Top End Health Department data which maintains yearly Indigenous and Non-Indigenous population data for the region. NT CDC data was used for confirmation of treatment history and serological titre response. NT CDC data for new cases of syphilis were divided into three categories during collection: Indigenous, non-Indigenous and unknown.

The Top End syphilis diagnostic algorithm uses serum TPPA treponemal testing as the initial test for syphilis and if positive, an RPR is done. CSF for suspected NS patients is analysed by VDRL, TPPA and FTA-A alongside biochemistry analysis and microscopy.

**Table 2. Clinical and laboratory characteristics based on 2014 US CDC diagnostic criteria.**

| | Definite | Probable | Total |
|---|---|---|---|
| | 9 | 16 | 25 |
| Male | 8 | 11 | 19 |
| Indigenous | 3 | 9 | 11 |
| HIV | 2 | 0 | 2 |
| **Clinical Diagnoses** | n = 8 | n = 16 | n = 24 |
| Meningitis | 0 | 0 | 0 |
| Ocular | 1 | 0 | 1 |
| Meningovascular | 1 | 1 | 2 |
| Dementia | 4 | 10 | 14 |
| Psychosis | 1 | 2 | 3 |
| Tabes Dorsalis | 1 | 2 | 3 |
| Seizures | 2 | 2 | 4 |
| **Laboratory** | | | |
| Number of reactive serum RPR | 9 | 10 | 19 |
| Median RPR titre | 1:128 (1:2–1:512) | 1:4 (1:1–1:16) | 1:8 (1:1–1:512) |
| Number of CSF examinations | 9 | 16 | 25 |
| Number of positive CSF VDRL ($\geq$ 1:1) | 9 | 0 | 9 |
| Number of reactive CSF TPPA | 8 | 2 | 10 |
| Number of reactive CSF FTA | 8 | 4 | 12 |
| Number with high WCC in CSF (WCC > 5 cell/microL) | 5 (median 26cell/microL, range 7-45cell/microL) | 2 (range 10-21cell/microL) | 7 (median 21 cell/microL, range 7–45 cell/microL) |
| Number with high CSF Protein (Protein > 0.50 mg/L) | 6 (median 1.08mg/L, range 0.68–1.37mg/L) | 16 (median 0.725mg/L, range 0.57–1.37mg/L) | 22 (median 0.77mg/L, range 0.57–1.37mg/L) |
| **Management** | n = 9 | n = 16 | n = 25 |
| Treatment for NS | 9 | 6 | 15 |

## Statistical analysis

All statistical analyses and graphs were generated using Graphpad Prism(v7, San Diego, California). Normally distributed data were compared using Student's t-test, and non-parametric comparisons were made using the Mann-Whitney U test. Proportions were examined using $\chi^2$ with Yates' correction or Fisher's exact test. Correlations were assessed using the Pearson test for correlated proportions for normal distributed variables and the Spearman rank test for non-normal distributed variables. Rate ratios and 95% confidence intervals were determined using Confidence Interval Analysis (vs 2.2.0).

## Ethics

Ethical approval was granted by the Human Research Ethics Committee of the Northern Territory Department of Health and Menzies School of Health Research (HREC 2016–2633)

## Results

Between January 2007 and December 2016, 3126 (1.6%) individuals in the Top End tested for syphilis serology were TPPA positive. Of these there were 1135 confirmed new cases of syphilis (Fig 1), of whom 789 (69.5%) were in Indigenous, 330 (29%) in non-Indigenous individuals and 18 (1.5%) had no classification available. The corresponding incidence of new cases of syphilis was 162.24 per 100,000 person years (py) [95% CI 151.12–173.97] in the Indigenous population and 24.10 per 100,00py [95% CI 21.57–26.85] in the non-Indigenous population

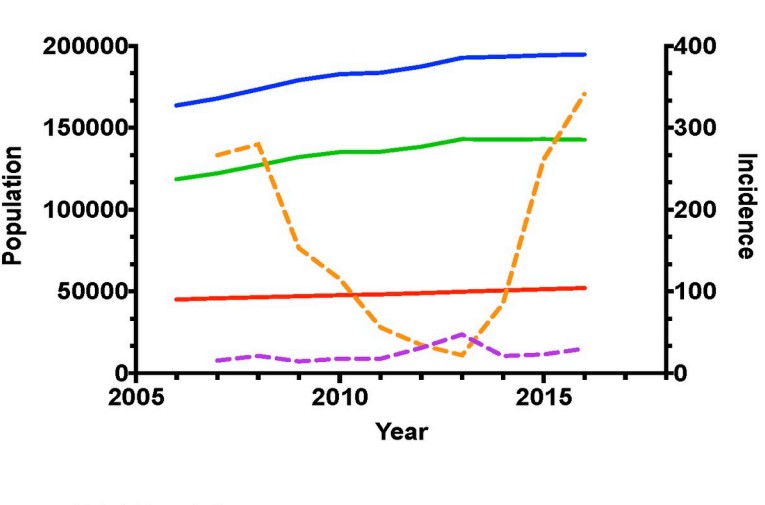

**Fig 1. Incidence of syphilis in the top end of the Northern Territory by Indigenous status; 2006–2016.**

(Rate Ratio (RR) = 6.73 [95%CI 5.92–7.65]; p<0.0001). The number of new cases rose from a mean of 105.8 per year between 2007 and 2011 to 121.2 per year between 2012 and 2016. The overall incidence of newly diagnosed syphilis cases was 61.16 [95% CI 57.66–64.83] per 100,000py.

## Neurosyphilis

Of the 3126 TPPA positive individuals, 75 (2.4%) underwent lumbar puncture. Of the 44 (59%) patients with CSF abnormalities, 16/44 (31%) had an alternate diagnosis or a history that was not suggestive of NS. Of the remaining 28 patients with clinical history consistent with NS 9 (32%) had a definite diagnosis of NS, 16 (57%) had a probable diagnosis of NS and 3 (11%) had possible NS. The nineteen patients with alternate diagnosis and possible NS were excluded from further analysis (Fig 2).

Of the 25 patients with definite or probable NS, 19 (76%) were male, 12 (48%) were Indigenous and 2(8%) were HIV positive (Table 3). The median age of these patients was 64 years (range: 32 to 97). The overall incidence of definite or probable NS was 1.37 [95% CI 0.87–1.99] per 100,000py, the incidence being 2.47 [95%CI 1.28–4.31] per 100 000py in the Indigenous population and 0.95 [95%CI 0.50–1.62] per 100,000py in the non-Indigenous population (rate ratio = 2.60 [1.19–5.70]; p = 0.017). When only patients with definite NS were included, the incidence was 0.49 [95% CI 0.22–0.92] per 100,000py, with no significant difference between the Indigenous (0.62 [95% CI 0.13–1.80] per 100,000py) and non-Indigenous (0.44 [95% CI 0.16–0.95] per 100,000py) populations p = 0.63. The rate of NS per case of syphilis in the Indigenous population was 12/789 (1.52%) and 13/330 (3.6%) (rate ratio = 0.39 [0.18–0.84]; p = 0.016).

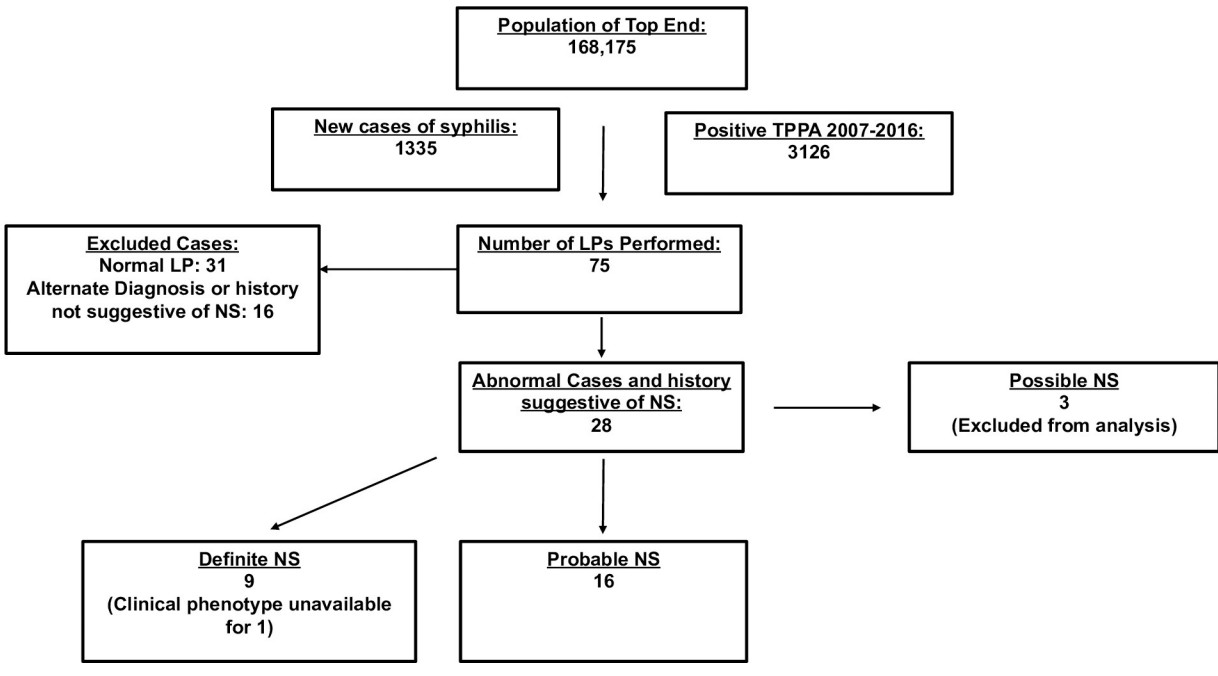

**Fig 2. Inclusion workflow for cohort.**

## Clinical manifestations

Clinical data were available for 24 of the 25 patients with definite or probable NS. Dementia was present in 14 (58.3%) of patients, seizures in 4 (16.7%), tabes dorsalis in 3 (12.5%), psychosis in 3 (12.5%), meningovascular disease in 2 (8.3%) and ocular involvement (panuveitis) in 1 (4.2%) patient. Three patients had more than one clinical manifestation of NS; Fig 3 and Table 2. There were no cases of meningitis or otic NS. It was not possible to characterise the dementia syndrome or severity from the clinical records. Overall 79% (19/24) cases were late manifestations of NS and 62% (5/8) of definite cases were late manifestations of the disease. No patients with asymptomatic NS were identified.

## Laboratory results

CSF VDRL was positive in all cases of definite NS and absent otherwise. CSF TPPA was positive in 89% (8/9) of those with definite NS and 18.8% (3/16) with probable NS. CSF FTA was only tested in 20 patients and was positive in all (8/8) patients with a definite diagnosis and 33% (4/12) of patients with a probable diagnosis. Only 2 patients in the probable group had positive CSF serology for both TPPA and FTA (Table 2).

**Table 3. Incidence of NS in the NT based on different criteria.**

|  | US CDC 2014 | US CDC 2018 |
|---|---|---|
| Total Incidence- per 100,000 py [95% CI] | 1.37 [0.87–1.99] | 1.02 [0.62–1.60] |
| Indigenous Incidence- per 100,000 py [95% CI] | 2.47 [1.28–4.31] | 1.85 [0.85–3.51] |
| Non-Indigenous Incidence—per 100,000 py [95% CI] | 0.95 [0.50–1.62] | 0.73 [0.35–1.34] |
| Rate Ratio Indigenous vs Non-Indigenous incidence [95% CI] | 2.60 [1.19–5.70] | 2.53 [1.03 to 6.24] |
|  | p = 0.017 | p = 0.043 |

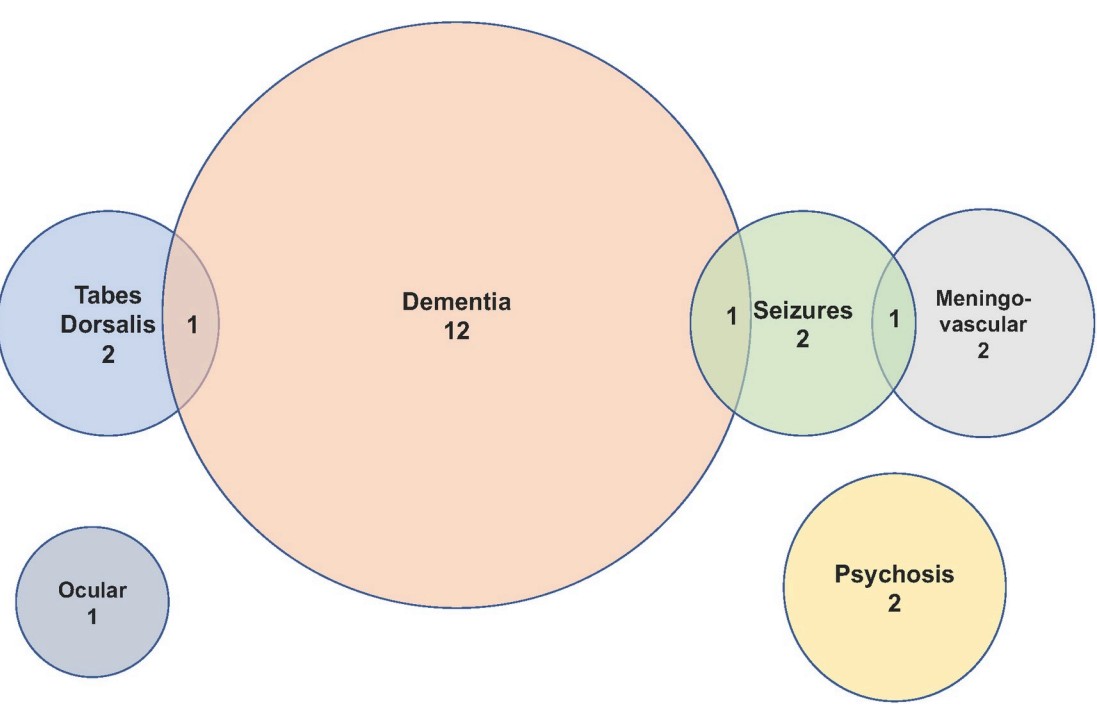

**Fig 3. Clinical syndromes and overlap.**

In total 28.0% (7/25) patients with definite or probable NS had a raised WCC in their CSF with a median count of 21 cell/microL (range 7 to 45 cell/microL), which was predominantly lymphocytic. In patients with a definite NS diagnosis WCC was elevated in 55.6% (5/9) of cases, compared to 12.5% (2/16) in patients with a probable diagnosis; p = 0.144. The CSF protein was elevated in 92% (24/25) of patients (median 0.74 mg/L, range 0.49–1.37mg/L).

Two patients with definite NS had borderline CSF results: with a CSF protein of 0.49mg/L and 0.5mg/L respectively with no pleocytosis. One patient with definite NS had a normal CSF protein with CSF WCC of 5 cell/microL which should be considered as a normal CSF. Two of these patients had dementia, there were no clinical data available for the third.

Serum RPR was reactive in all (9/9) patients with a definite diagnosis of NS and 62.5% (10/16) of those with a probable diagnosis, with a significantly higher titre in those with definite NS; p<0.0001 (Table 2).

## Treatment

Prior to a diagnosis of NS, two patients with definite NS had been treated for late latent syphilis with intramuscular benzathine penicillin without a pertinent four-fold decrease in their RPR. Nine (56.3%) of the patients with probable NS had also received prior intramuscular benzathine penicillin for latent syphilis of whom 4 (44.4%) did not have an appropriate four-fold decrease in their RPR titre. This can be attributed to either treatment failure or potential re-infection. Of the 6 patients in the probable group who had a non-reactive serum RPR at the time of NS diagnosis, data regarding prior treatment through the NT CDC was available for only 5. Of those 5 patients, 3 had no documented prior history of treatment for syphilis.

All patients with definitive NS were treated with the recommended regimen of 14–15 days of IV benzylpenicillin, but only 37.5% (6/16) of those with probable NS were treated with IV benzyl penicillin. Of the 10 patients who were not treated appropriately, four were treated for late latent syphilis (weekly IM benzathine penicillin for 3 weeks), three of whom had dementia

with the only mildly elevated CSF protein (range 0.57–0.74 mg/L). One patient treated for latent syphilis had signs and symptoms of tabes dorsalis and a CSF WCC 21 cell/microL and a negative CSF VDRL.

Only 2 of the 25 patients with definite or probable NS had a repeat lumbar puncture, one of whom was a 60-year-old who had presented with psychosis and received appropriate therapy for NS, but subsequently required re-treatment as there was still an elevated CSF VDRL without a 4 fold decrease.

## Discussion

Despite significant public health endeavours, syphilis continues to increase in incidence and exert significant morbidity in at risk populations. Our analysis highlights that between 2007 and 2016, the incidence of syphilis in the Top End of Australia was 61.16 per 100,000 py with an associated incidence of NS, based on the 2014 US CDC criteria, of 1.37 per 100,000 py. The rate of syphilis was almost 7-fold higher in the Indigenous population than in the non-Indigenous population and this was associated with a 2.5 fold higher rate of NS. However, the rate of NS per case of syphilis was significantly lower in the Indigenous population in comparison. This is likely due to the stringent follow up and treatment by the NT CDC of Indigenous cases compared to non-Indigenous cases. This has likely limited the duration of syphilis exposure in this community and prevented the development of NS.

Since two thirds of the cases were defined as probable diagnoses we may have overestimated the true risk of NS however even when these patients were excluded the incidence of definitive NS in the Top End was 0.49 per 100,000py, approximately 2.5–5 fold higher than that reported from UK and Denmark. We may also have underestimated the true incidence. Our methodology screened patients based on those who underwent lumbar punctures, all of whom had presented with neurological symptoms. This approach would have failed to capture asymptomatic NS patients, and thus underestimated the true incidence. It is not current practice to perform an LP for patients with syphilis who do not have a four-fold decline in serum RPR despite adequate treatment and this may have missed detection of asymptomatic cases. Our proposed incidence of NS also assumes that clinicians have considered the diagnosis of NS in all patients presenting with suspicious symptoms and sent appropriate testing.

In view of the rising number of cases of early syphilis it is likely that NS will increase over the coming decades. Our study is unique in applying almost complete capture of reported cases from the region. It is also the first study to detail the incidence and clinical characteristics of NS in any region of Australia. Similar to previous studies in the US that demonstrate racial disparities with the rates of syphilis between Whites, Blacks and Hispanic persons, our study demonstrates marked ethnic disparities in the incidence of both syphilis and neurosyphilis [7, 18], which further undermines the already poor health outcomes and lower life expectancy of Indigenous Australians [18].

The diagnosis of NS is challenging, since there is no gold standard microbiological assay and varying, non-specific diagnostic criteria. Alarmingly more than half of the patients in our study with a probable diagnosis of NS did not receive appropriate treatment. Both false negative and false positive diagnoses contribute to the misdiagnosis of NS. The former arises from the misconception that a negative CSF VDRL has a high negative predictive value for ruling out NS. Whilst CSF VDRL is highly specific, its sensitivity is between (30 and 70%) [19]. A negative CSF VDRL should therefore not dissuade a clinician from the potential diagnosis of NS. Although CSF treponemal tests have higher sensitivity, their negative predictive value is dependent on the pre-test probability of NS [17]. The high prevalence of syphilis in the Top End and the high level of suspicion prompting an LP and investigation for NS increases the pre-test probability of NS in our cohort. In this context a negative CSF treponemal test cannot

be relied upon to rule out NS. Although polymerase chain reaction (PCR) carries a high sensitivity and specificity for other specific neuroinfectious diseases (HSV, enterovirus, etc), the utility of PCR has never been shown for NS [20].

False positive diagnoses of NS can arise from the low specificity and high sensitivity of the US CDC 2014 diagnostic criteria, in which the only CSF abnormality required for a diagnosis of NS is an elevated CSF protein. US CDC Guidelines recommend that NS should be treated with intravenous benzylpenicillin administered 4 hourly for 15 days [14]. We conjecture that there is either a reluctance to treat an already very demented patient or ambivalence over the significance of a mildly elevated protein as the only abnormal CSF finding. The clinical experience of many clinicians is that a mildly elevated CSF protein (0.46–0.55 mg/L) as an isolated finding is of little clinical utility [21]. We were unable to extract data on the cognitive profile of our patients with dementia and it is therefore not possible to differentiate between patients with probable NS who truly had dementia related to NS or due to another aetiology. Given the increasing prevalence of dementia in the general community, the treating clinicians may have decided not to treat for probable NS as another cause of dementia was more likely.

During the preparation of the manuscript, the US CDC released revised diagnostic criteria which included an additional requirement that a reactive serum RPR result was needed for a diagnosis of definite and probable NS. A four-fold decrease in RPR titres suggests a treatment response in syphilis, however RPR titres can fall and normalise even without treatment. Previous studies have demonstrated that in patients with definite NS, the serum RPR is almost always reactive, but approximately 10% of patients with probable NS have a non-reactive serum RPR [22]. In our cohort all patients with definite NS had a reactive serum RPR, but over a third of patients with probable NS group did not have a reactive serum, of whom three had no history of prior syphilis treatment and only had an elevated CSF protein. According to more stringent criteria these serum RPR negative patients would be considered false positives. Although some of these patients may have been correctly categorized as not having NS, the diagnosis in those who have never received any form of prior syphilis treatment is unclear–they could have potentially normalised serum RPR overtime and thus constitute false negative diagnosis of NS. This group is likely to constitute a very small overall percentage of NS cases and a drop in the US CDC 2018 criteria's sensitivity may be appropriate for the compensatory increase in specificity. The US CDC 2018 criteria does demonstrate that a large number that were previously categorized as probable NS with only an elevated CSF protein, likely did not have NS as they were no longer considered probable under the new criteria. We recategorized our cohort into both the 2014 and 2018 NS criteria, (Fig 4, S1 File) to demonstrate the differences in incidence between the two criteria. With the new 2018 CDC NS criteria, only 3/10 (30%) of probable NS cases were treated appropriately with IV benzylpenicillin for 2 weeks. 3 patients received 3 courses of IM benzathine penicillin and the remainder of patients received no treatment.

Our study has several limitations. Although we attempted to capture all cases of syphilis and NS searching multiple databases, given the uncertainty of the diagnosis and our reliance on LP and clinical suspicion, we may have either over- or under- estimated the true burden of disease. Furthermore whilst we included prospective data collection the majority of the data were gathered retrospectively, and we were unable to document detailed phenotypes of many of the patients from the medical notes.

## Conclusion

Our study highlights the ongoing difficulties with the diagnosis and management of NS. Despite these challenges, there continues to be significant disparities between Indigenous and non-Indigenous Australians in the incidence of syphilis and NS. Similar to other studies, the

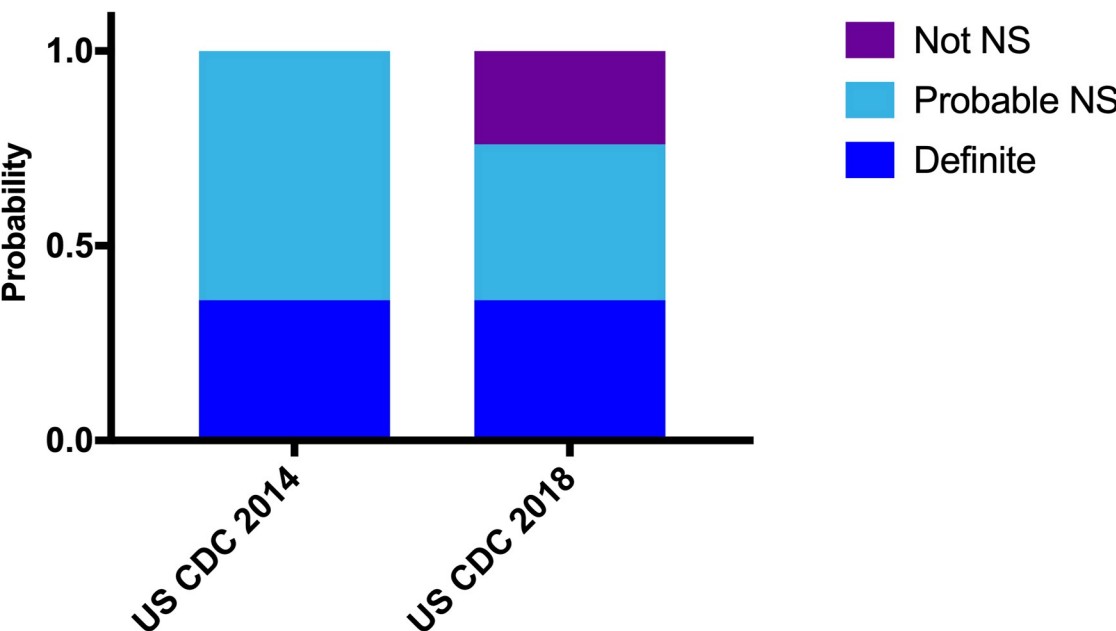

**Fig 4. Evaluation of patient cohort with different diagnostic criteria.** The percentage of Probable cases decreases from 64% to 40% respectively. The Category of not NS subsequently increases to 24% respectively.

key dilemma is whether patients with probable NS truly have the disease and would benefit from appropriate treatment [23]. Until the advent of a new and accurate assay, NS will continue to be a complex and difficult disease to diagnoses and manage. A large proportion of patients with probable NS were not diagnosed or treated appropriately. Whilst the new 2018 CDC NS criteria improves the specificity of probable NS, it is critical that clinicians should be aware of the diagnosis and treat patients appropriately.

## Supporting information

**S1 File.**
(DOCX)

## Author Contributions

**Conceptualization:** Prashanth S. Ramachandran, James N. Burrow, Ric N. Price.

**Data curation:** Prashanth S. Ramachandran, Rob W. Baird, Peter Markey.

**Formal analysis:** Prashanth S. Ramachandran, Ric N. Price.

**Methodology:** Prashanth S. Ramachandran, James N. Burrow, Ric N. Price.

**Supervision:** James N. Burrow, Ric N. Price.

**Writing – original draft:** Prashanth S. Ramachandran.

**Writing – review & editing:** Rob W. Baird, Peter Markey, Sally Singleton, Michael Lowe, Bart J. Currie, James N. Burrow, Ric N. Price.

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
