## [Decision Letter · Decision Letter 0]

14 Jul 2020

PONE-D-20-18019

Neurosyphilis: Still Prevalent and Overlooked in an At Risk Population

PLOS ONE

Thank you for submitting your manuscript to PLOS ONE. After careful consideration, we feel that it has merit but does not fully meet PLOS ONE’s publication criteria as it currently stands. Therefore, we invite you to submit a revised version of the manuscript that addresses the points raised during the review process.

We look forward to receiving your revised manuscript.

Kind regards,

Walter R. Taylor

Academic Editor

PLOS ONE

Additional Editor Comments:

Dear Dr. Ramachandran,

thank you for submitting this interesting paper on neurosyphilis.

We have received one review so far but I am happy to proceed with a revision of your paper.

As the academic editor, I would also like you to consider a couple of points.

The rate of lumbar puncture was 2.4% - do you know how this compares with other countries ?

Is there a take home message for practising clinicians, given the low rate of appropriate treatment ?

Could you add a line or two telling us what you think the limitations are of your study, please ?

yours sincerely,

Bob Taylor.

2. Please provide additional details regarding participant consent. In the ethics statement in the Methods and online submission information, please ensure that you have specified (1) whether consent was obtained from the participants of this study (2) whether consent was informed and (3) what type you obtained (for instance, written or verbal, and if verbal, how it was documented and witnessed). If the need for consent was waived by the ethics committee, please include this information.

3. Please include a summary table of collected patient demographics.

"No"

"No"

6. We note you have included a table to which you do not refer in the text of your manuscript. Please ensure that you refer to Table 3 in your text; if accepted, production will need this reference to link the reader to the Table.

Reviewers' comments:

Reviewer's Responses to Questions

**Comments to the Author**

1. Is the manuscript technically sound, and do the data support the conclusions?

Reviewer #1: Yes

2. Has the statistical analysis been performed appropriately and rigorously? 

Reviewer #1: N/A

3. Have the authors made all data underlying the findings in their manuscript fully available?

Reviewer #1: No

4. Is the manuscript presented in an intelligible fashion and written in standard English?

Reviewer #1: Yes

5. Review Comments to the Author

Reviewer #1: The authors review the experience with neurosyphilis in the North Territory of Australia, a territory that has a centralized syphilis registry, over a 10-year period (2007-2016). They identify 25 patients with neurosyphilis of whom 9 were definite and 16 probable based on 2014 U.S. CDC criteria. The important finding from the study is the failure to have treated 68% of the probable cases as they were CSF VDRL negative.

1) The authors estimate incidence rates in their population. While they comment on the risk of overestimate in their discussion due to the inclusion of probable cases, they should also comment on the possibility that these are underestimates (at least using the diagnostic criteria that have been employed). Neurosyphilis is often asymptomatic or not considered in the differential diagnosis. Unless the entire population was screened by anti-treponemal antibody, one really does not know the true prevalence of infection.

2) The authors state that the incidence of syphilis declined significantly following the introduction of penicillin. While this is true, contact tracing also played a significant role in the decline in incidence in the U.S. as case rates began falling long before the widespread institution of penicillin.

3) The increased incidence in the indigenous population, presumably individuals in a lower socioeconomic group relative to the white population, parallels the experience in the U.S. where rates of syphilis is substantially higher in the African-American and Hispanic populations.

4) Two of their patients were HIV-infected. Can they state what percentage of their patients were men who have sex with men among both the indigenous and non-indigenous populations in their study?

5) Were there not any asymptomatic neurosyphilis cases? If not, why not? Also, can the authors identify whether the neurosyphilis manifestations were early or late manifestations of the infection. Whereas dementia, tabes, and psychosis are typically late manifestations (tertiary syphilis), meningitis, seizures and uveitis are often seen with secondary syphilis, particularly, in the HIV infected population.

6) How comfortable are the authors in attributing the neurological manifestations to syphilis? After all, “a man can have as many diseases as he damn well pleases.” If the clinicians caring for these patients felt comfortable that there were better explanations for their neurological manifestations might that not explain why those with “probable neurosyphilis” were not treated?

7) “the test carries a high sensitivity and low sensitivity” (not “now”) line 129

8) How do the authors explain the apparent profound decline in syphilis in the indigenous population between 2007 and 2015 with a nadir in 2013, a year that the incidence in the non-indigenous increased?

9) How does the application of the 2018 CDC criteria change the numbers with respect to the percentage of probably neurosyphilis in which there was a failure to treat?

6. PLOS authors have the option to publish the peer review history of their article (what does this mean?). If published, this will include your full peer review and any attached files.

Reviewer #1: **Yes: **Joseph R. Berger, M.D.

---

## [Author Response · Author response to Decision Letter 0]

17 Aug 2020

The rate of lumbar puncture was 2.4% - do you know how this compares with other countries ?

Thank you for this excellent suggestion. Unfortunately, it is not clear after reviewing the literature what LP rates were in other countries. As we one of our methods of screening for NS patients was by reviewing all LP performed in the region we were able to arrive at those figures. Other studies have either determined NS through chart reviews, referrals to tertiary centers or reporting to their local CDC. These cases have already met criteria for NS. It is unclear from these studies how many patients had an LP for a query of NS but did not met criteria, and therefore not diagnosed with NS.

Is there a take home message for practicing clinicians, given the low rate of appropriate treatment ?

We have added a line to the conclusion to try address this. Line 386

Could you add a line or two telling us what you think the limitations are of your study, please ?

We have added a paragraph explaining the limitations of the study, including the difficulties in making a definitive diagnosis of NS and our reliance on mostly retrospective data. Line 373

The authors estimate incidence rates in their population. While they comment on the risk of overestimate in their discussion due to the inclusion of probable cases, they should also comment on the possibility that these are underestimates (at least using the diagnostic criteria that have been employed). Neurosyphilis is often asymptomatic or not considered in the differential diagnosis. Unless the entire population was screened by anti-treponemal antibody, one really does not know the true prevalence of infection.

We agree that this is an important point. We have now raised the issues in the discussion along with the comments regarding asymptomatic cases. This is on line 280 of the discussion. 

The authors state that the incidence of syphilis declined significantly following the introduction of penicillin. While this is true, contact tracing also played a significant role in the decline in incidence in the U.S. as case rates began falling long before the widespread institution of penicillin.

Thank you for this addition and something we have now mentioned with appropriate referencing on line 90

The increased incidence in the indigenous population, presumably individuals in a lower socioeconomic group relative to the white population, parallels the experience in the U.S. where rates of syphilis is substantially higher in the African-American and Hispanic populations.

Yes, this is indeed true, the Australian Indigenous population are individuals who have a considerably lower socioeconomic background compared to Whites, with poorer health outcomes. We have added an additional line regarding this in our discussion (line 313). 

Two of their patients were HIV-infected. Can they state what percentage of their patients were men who have sex with men among both the indigenous and non-indigenous populations in their study?

Unfortunately, we do not have that data as it was not consistently or clearly documented in the medical records. It was not available to us from the data provided from the CDC either. 

Were there not any asymptomatic neurosyphilis cases? If not, why not? Also, can the authors identify whether the neurosyphilis manifestations were early or late manifestations of the infection. Whereas dementia, tabes, and psychosis are typically late manifestations (tertiary syphilis), meningitis, seizures and uveitis are often seen with secondary syphilis, particularly, in the HIV infected population.

There were no cases of asymptomatic NS, which was likely due to our methodology of screening cases based on LPs performed, all of who had neurological symptoms. It also likely relates to the practice of not consistently testing for NS with an LP in patients with syphilis that fail to have a four-fold decrease in RPR despite treatment. There is probably a much higher prevalence of asymptomatic cases that we have failed to detect. This has been added to the discussion (line 282) as well as the results section (line 222). We have categorized cases into early and late manifestations and added it to the results section (line 220), 79% of cases were late manifestations. 

How comfortable are the authors in attributing the neurological manifestations to syphilis? After all, “a man can have as many diseases as he damn well pleases.” If the clinicians caring for these patients felt comfortable that there were better explanations for their neurological manifestations might that not explain why those with “probable neurosyphilis” were not treated?

This is an excellent point that was raised amongst our group. We have added our thoughts regarding this in the discussion (line 340), specifically around the issue of dementia, which is a common condition in the general population. If the clinician felt that the cause of dementia was due to a more common or likely cause then they may not have treated a diagnosis of probable NS. As we do not have in-depth cognitive testing results on our patient cohort, it is difficult for us to say that there was not another dementing syndrome. Further prospective studies are planned to address this.

“the test carries a high sensitivity and low sensitivity” (not “now”) line 129

Thank you, correction made

How do the authors explain the apparent profound decline in syphilis in the indigenous population between 2007 and 2015 with a nadir in 2013, a year that the incidence in the non-indigenous increased?

The vast majority of cases between 2007-2015 were late latent cases of syphilis in both the Indigenous and non-Indigenous population. These cases were on the decline in the indigenous population, which we believe was in part due to the work by the CDC, though this is difficult to say in retrospect. The small rise in 2013 in the non-indigenous population is in late latent cases as well, though we are not sure why there was a rise. The sharp increase in cases from 2013 onwards are all new cases of syphilis in the Indigenous population, with late latent cases remaining stable in both groups. We have added a table with a breakdown of the raw numbers as a supplementary table for additional clarity. 

How does the application of the 2018 CDC criteria change the numbers with respect to the percentage of probably neurosyphilis in which there was a failure to treat?

Thank you for this suggestion as it was not something we had thought to look at. In fact with the 2018 CDC criteria only 3/10 cases received IV BP and 3 cases received IM benzathine. This is less than with the 2014 criteria. We have added these findings to the abstract and discussion section (line 74 & 369).

---

## [Editor Report · Decision Letter 1]

21 Aug 2020

Neurosyphilis: Still Prevalent and Overlooked in an At Risk Population

PONE-D-20-18019R1

Dear Dr.Ramachandran,

We’re pleased to inform you that your manuscript has been judged scientifically suitable for publication and will be formally accepted for publication once it meets all outstanding technical requirements.

Kind regards,

Walter R. Taylor

Academic Editor

PLOS ONE

Additional Editor Comments (optional):

I noticed a handful of small errors - spelling and punctuation. Please see the attached.
---

## [Editor Report · Acceptance letter]

24 Sep 2020

PONE-D-20-18019R1 

Neurosyphilis: Still Prevalent and Overlooked in an At Risk Population 

Dear Dr. Ramachandran:

I'm pleased to inform you that your manuscript has been deemed suitable for publication in PLOS ONE. Congratulations! Your manuscript is now with our production department. 

Kind regards, 

on behalf of

Dr. Walter R. Taylor 

Academic Editor

PLOS ONE